# Studying mass generation for gluons

**Gernot Eichmann[1,2⋆] and Jan M. Pawlowski[3,4]**

**1** LIP Lisboa, Av. Prof. Gama Pinto 2, 1649-003 Lisboa, Portugal
**2** Departamento de Física, Instituto Superior Técnico, 1049-001 Lisboa, Portugal
**3** ExtreMe Matter Institute EMMI, GSI, Planckstr. 1, 64291 Darmstadt, Germany
**4** Institut für Theoretische Physik, Universität Heidelberg,
Philosophenweg 16, 69120 Heidelberg, Germany

⋆ gernot.eichmann@tecnico.ulisboa.pt

## Abstract

In covariant gauges, the gluonic mass gap in Yang-Mills theory manifests itself in the basic observation that the massless pole in the perturbative gluon propagator disappears in nonperturbative calculations, but the origin of this behavior is not yet fully understood. We summarize a recent study of the respective dynamics with Dyson-Schwinger equations in Landau-gauge Yang-Mills theory. We identify the parameter that distinguishes the massive Yang-Mills regime from the massless decoupling solutions, whose endpoint is the scaling solution. Similar to the PT-BFM scheme, we find evidence that mass generation in the transverse sector is triggered by longitudinal massless poles.

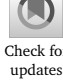

## 1 Introduction

The origin of mass continues to be one of the outstanding questions in QCD and hadron physics. While mass generation in the quark sector can be tied to the dynamical breaking of chiral symmetry, see e.g. [1–4] and references therein, the underlying mechanism in the Yang-Mills sector of QCD is not yet fully understood. The emergence of a mass gap and the related question of confinement must be encoded in the $n$-point correlation functions of pure Yang-Mills theory, whose basic representative is the gluon propagator

$$D^{\mu\nu}(Q) = \frac{1}{Q^2}\left(Z(Q^2)\,T_Q^{\mu\nu} + \xi\,L(Q^2)\,L_Q^{\mu\nu}\right).\qquad(1)$$

Here, $T_Q^{\mu\nu} = \delta^{\mu\nu} - Q^\mu Q^\nu/Q^2$ and $L_Q^{\mu\nu} = Q^\mu Q^\nu/Q^2$ are the transverse and longitudinal projection operators with respect to the four-momentum $Q^\mu$, $\xi$ is the gauge parameter, and $\xi = 0$

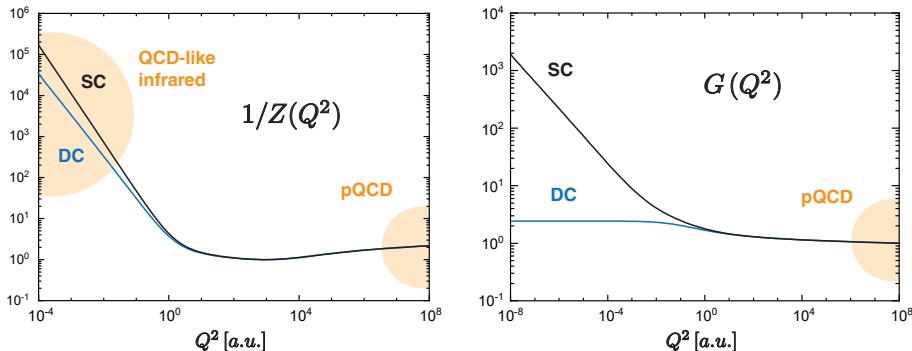

Figure 1: DSE solutions for the inverse gluon dressing function $1/Z(Q^2)$ and ghost dressing function $G(Q^2)$, where 'DC' denotes decoupling and 'SC' scaling [5].

corresponds to Landau gauge. In linear covariant gauges, the longitudinal dressing function $L(Q^2) = 1$ is trivial due to gauge invariance. The transverse gluon dressing function $Z(Q^2)$ is the quantity of interest in what follows.

The basic feature of gluon mass generation is the disappearance of the massless pole in the perturbative gluon propagator in nonperturbative calculations, where $Z(Q^2)/Q^2$ does not diverge at the origin. This is a robust outcome of lattice calculations [6–10] and functional methods such as Dyson-Schwinger equations (DSEs) and the functional renormalization group (fRG) [11–24]. What happens instead is still an open question, as the propagator could develop poles at timelike momenta, complex conjugate poles, branch cuts, etc. In any case, the absence of a massless pole implies $Z(Q^2 \to 0) = 0$, and therefore $1/Z(Q^2)$ must have a massless singularity at the origin as shown in Fig. 1. But how does this nonperturbative singularity arise in the first place?

The infrared behavior seen on the lattice corresponds to the *massive* or *decoupling* solution obtained with DSE and fRG calculations in QCD [16–21] and the Curci-Ferrari model [22], where the gluon propagator freezes out in the infrared and $1/Z(Q^2 \to 0) \propto 1/Q^2$. In turn, the ghost dressing function $G(Q^2 \to 0)$ becomes constant as shown in Fig. 1. Because $1/Z(Q^2)$ is the solution of the gluon DSE, which is an exact equation, at least one of the diagrams in the DSE (cf. Fig. 2) must develop a massless $1/Q^2$ singularity. In the Pinch-Technique/Background-Field method (PT-BFM), which allows one to rearrange the diagrams in the DSE according to gauge invariance, mass generation is triggered by massless longitudinal poles in the three-gluon vertex [20, 25–27]. Does this also happen in the standard treatment of the DSEs in Landau gauge?

Moreover, calculations with functional methods also find the *scaling* solution, where the $n$-point correlation functions scale with infrared power laws [11–15]. For the gluon and ghost dressing functions this entails $Z(Q^2 \to 0) \propto (Q^2)^{2\kappa}$ and $G(Q^2 \to 0) \propto (Q^2)^{-\kappa}$ with an infrared exponent $\kappa \approx 0.595$, which is also plotted in Fig. 1. Here the inverse gluon dressing diverges slightly faster than a $1/Q^2$ pole and also the ghost dressing is infrared-divergent. The scaling solution is consistent with the Kugo-Ojima confinement scenario based on global BRST symmetry [18, 28], and it leads to a $1/Q^4$ behavior and thus a linear rise in coordinate space for the (gauge-dependent) quark-antiquark four-point function [29], whose gauge-invariant version defines the Wilson loop.

Functional calculations have further revealed a family of decoupling solutions with the scaling solution as their endpoint [17, 18, 22]. While the scaling solution is not seen on the lattice, there are indications for different decoupling solutions depending on the gauge-fixing procedure [30]. It has been speculated that the emergence of a family of solutions may be due to an additional gauge fixing parameter in Landau gauge [31]. Indeed, there is accumulating

Figure 2: Coupled Yang-Mills DSEs for the ghost propagator, gluon propagator and three-gluon vertex.

evidence that all these solutions may be physically equivalent, as physical observables agree within the systematic error bars: e.g., the Polyakov loop expectation value, which is the order parameter for center symmetry in Yang-Mills theory, vanishes for scaling *and* decoupling-type solutions alike and thus implies confinement for both [32,33]. Moreover, the respective critical temperature agrees within the error bars for the whole set of scaling and decoupling solutions, while generally it depends on the mass in massive Yang-Mills theory. This begs the question: What is the parameter that distinguishes these different types of solutions?

## 2 Yang-Mills DSEs

To answer these questions, we solve the coupled system of DSEs for the ghost propagator, gluon propagator and three-gluon vertex in Fig. 2; see Ref. [5] for details of the calculation. Here, we take all diagrams in the ghost and gluon equations into account, whereas in the three-gluon vertex DSE we neglect diagrams with two-loop terms and higher $n$-point functions. In addition, we restrict ourselves to the leading tensor of the three-gluon vertex in the symmetric limit, which is a good approximation in Landau gauge [34]. Thus, the quantities we compute are the gluon dressing $Z(Q^2)$, the ghost dressing $G(Q^2)$ and the three-gluon vertex dressing $F_{3g}(Q^2)$. The remaining inputs are the ghost-gluon vertex, which is kept at tree level, and the four-gluon vertex, whose tree-level tensor is multiplied with $F_{4g}(Q^2) = G(Q^2)^2/Z(Q^2)$ and updated dynamically during the iteration. Similar high-quality truncations (also including higher $n$-point functions) have been employed in DSE and fRG calculations [21,23].

We emphasize that different truncations do not change the qualitative features we discuss in the following, which also remain intact when considering only the ghost and gluon DSEs as in Refs. [12–14]. The Slavnov-Taylor identities (STIs) provide an internal way to quantify the truncation error, which is about 10% when neglecting the two-loop terms in the gluon DSE and $3-4\%$ when solving the full system in Fig. 2.

To study the origin of mass generation, we decompose the gluon self-energy in the second row of Fig. 2 in the following overcomplete basis:

$$\Pi^{\mu\nu}(Q) = \Delta_T(Q^2)(Q^2\,\delta^{\mu\nu} - Q^\mu Q^\nu) + \Delta_0(Q^2)\,\delta^{\mu\nu} + \Delta_L(Q^2)Q^\mu Q^\nu. \qquad (2)$$

This allows us to isolate quadratic divergences, which can only arise in the term $\Delta_0(Q^2)$ due to the hard momentum cutoff employed in the equations and must be subtracted. The logarithmic divergences in the remaining terms are absorbed in the standard renormalization. The projection of the gluon DSE onto its Lorentz-invariant components yields

$$Z(Q^2)^{-1} = Z_A + \Delta_T(Q^2) + \frac{\Delta_0(Q^2)}{Q^2}, \qquad L(Q^2)^{-1} = 1 + \xi\left(\Delta_L(Q^2) + \frac{\Delta_0(Q^2)}{Q^2}\right) \qquad (3)$$

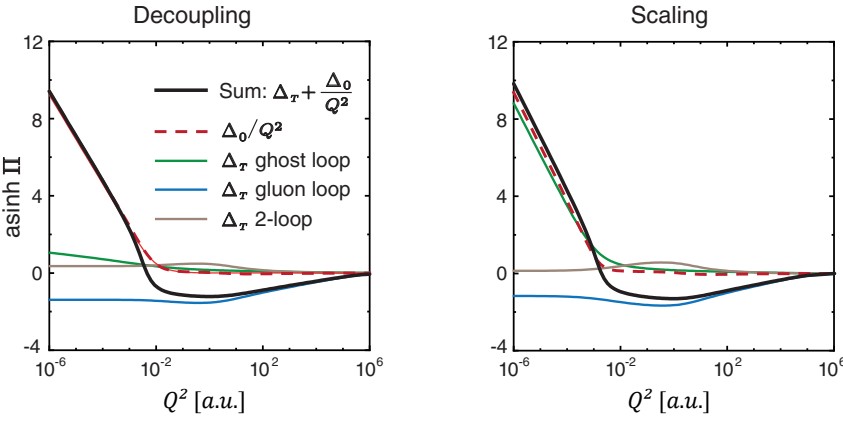

Figure 3: Gluon self-energy contributions for the decoupling and scaling case [5].

for the transverse and longitudinal dressing functions, where $Z_A$ is the gluon renormalization constant. The STI for the gluon propagator demands that the self-energy must be completely transverse, which leaves two possible options:

$$\text{Scenario A: } \Delta_L = \Delta_0 = 0, \qquad \text{Scenario B: } \Delta_L = -\frac{\Delta_0}{Q^2}. \qquad (4)$$

Thus, $\Delta_L$ and $\Delta_0$ must either vanish identically after removing the quadratic divergences, or there must be a cancellation between them.

From the resulting self-energy contributions in Fig. 3, one can clearly see that $\Delta_0$ is non-zero. In fact, for the decoupling solutions $\Delta_0$ is the term responsible for mass generation as it enters like $1/Q^2$. The ghost loop contribution to $\Delta_T$ diverges logarithmically, whereas all other contributions become constant in the infrared; also $\Delta_0$ goes to a constant. By contrast, for the scaling solution both $\Delta_T$ and $\Delta_0/Q^2$ diverge with the same power $1/(Q^2)^{2\kappa}$ in the infrared which originates from the ghost loop.

In Scenario A, a nonzero term $\Delta_0$ can at best be an artifact, either from the truncation or from the hard cutoff. One way to proceed is then to replace the dynamically calculated $\Delta_0$ by a constant, which yields a mass term like in massive Yang-Mills theory, and send $\Delta_0 \to 0$ in the end. This only leaves the scaling solution (as one can already infer from Fig. 3), however with an ambiguity in the infrared exponent $\kappa$; a similar ambiguity arises when determining $\kappa$ analytically [12, 13]. Based on these observations, our analysis disfavors Scenario A.

In Scenario B, on the other hand, the longitudinal consistency relation (4) between $\Delta_0$ and $\Delta_L$ does not affect the transverse equation and the $\Delta_0/Q^2$ term, but it implies that $\Delta_L$ must have a massless $1/Q^2$ pole. From the self-energy in Eq. (2) one infers that this can only happen if either of the vertices (the ghost-gluon vertex, three-gluon vertex or four-gluon vertex) has a longitudinal massless pole. How can one find out?

Let us first investigate what distinguishes the scaling and decoupling solutions. Usually this is implemented by a boundary condition on the ghost: After setting renormalization conditions, the Yang-Mills equations depend on the gluon dressing $Z(\mu^2)$ at some renormalization scale $\mu$, the ghost dressing $G(\nu^2)$, and the coupling $g$. Without loss of generality one can renormalize the ghost at $\nu^2 = 0$, so that $Z(\mu^2)$, $G(0)$ and $g$ enter in the equations. If $g$ and $Z(\mu^2)$ are kept fixed and $G(0)$ is varied, this leads to the family of decoupling solutions ($G(0)$ finite) with the scaling solution as their endpoint ($G(0) \to \infty$). From the viewpoint of renormalization, this is however not completely satisfactory as $G(0)$ should only renormalize the ghost propagator but not lead to different solutions. This is also seen in fRG computations, e.g. [21], which are manifestly RG-consistent.

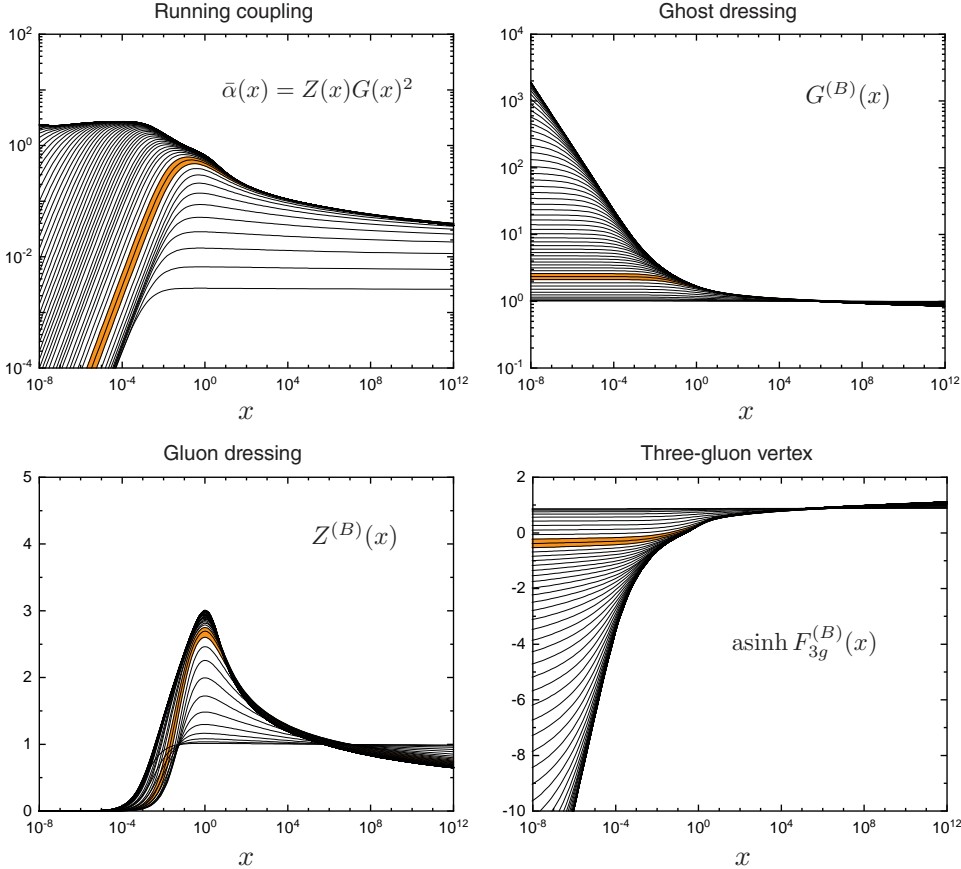

Figure 4: Solutions for the running coupling and the ghost, gluon and three-gluon vertex dressing functions at a fixed value of $\beta$ and varying $\alpha$. [5].

To this end, one observes that the arbitrariness in the subtraction of the quadratic divergences in $\Delta_0(Q^2)$ may be compensated by a parameter $\beta$:

$$\frac{\Delta_0(Q^2)}{Q^2} \rightarrow \frac{\Delta_0(Q^2) - \Delta_0(Q_0^2)}{Q^2} + \frac{g^2}{4\pi} G(0)^2 \beta \frac{\mu^2}{Q^2}. \tag{5}$$

This introduces an effective mass term in the equations (the prefactors ensure the correct renormalization), which therefore depend on an *additional* parameter $\beta$. In Scenario A mentioned above, the first term in Eq. (5) is dropped and $\beta$ is sent to zero in the end, whereas in Scenario B this term is dynamical and $\beta$ remains. It also turns out that $Z(\mu^2)$, $G(0)$ and $g$ are not actually independent but only appear in the equations through the combination

$$\alpha = \frac{g^2}{4\pi} Z(\mu^2) G(0)^2 \in \mathbb{R}_+. \tag{6}$$

This can be seen by redefining $Z(Q^2) \rightarrow Z(Q^2)/Z(\mu^2)$, $G(Q^2) \rightarrow G(Q^2)/G(0)$ and performing the same operations for the three- and four gluon vertex as well as the renormalization constants. Moreover, when introducing a dimensionless scale $x = Q^2/(\beta\mu^2)$ and redefining $G(x) \rightarrow \sqrt{\alpha} G(x)$, the equations for the ghost and gluon dressing functions assume the compact form

$$G(x)^{-1} = \frac{1}{\sqrt{\alpha}} + \Sigma(x) - \Sigma(0), \qquad Z(x)^{-1} = 1 + \Pi(x) - \Pi(\tfrac{1}{\beta}). \tag{7}$$

Here, $\Sigma(x)$ is the ghost self-energy and $\Pi(x)$ is the transverse part of the gluon self-energy in Eq. (3) including the $\Delta_T$ and $\Delta_0$ terms. Only $\alpha$ and $\beta$ appear explicitly in the equations, which allows one to study the behavior of the solutions in the $(\alpha, \beta)$ plane.

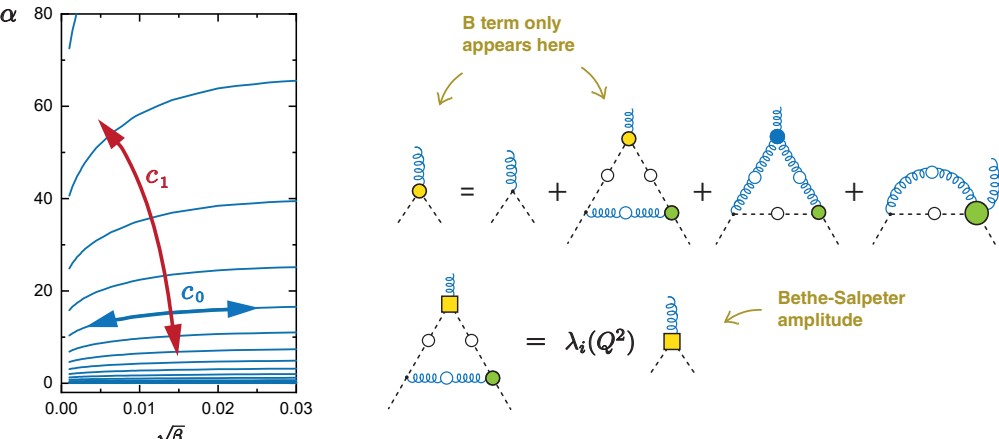

Figure 5: Left: Lines of constant physics in the $(\alpha, \beta)$ plane [5]. Right: DSE for the ghost-gluon vertex, which becomes a homogeneous BSE for the longitudinal $B$ term.

Fig. 4 displays the solutions for a fixed value of $\beta$ and varying $0 < \alpha < \infty$. The family of decoupling solutions is characterized by the parameter $\alpha$, and the scaling solution is the envelope of the decoupling solutions for $\alpha \to \infty$. The running coupling $\bar{\alpha}(x) = Z(x) G(x)^2$ is renormalization-group invariant; this is the quantity that sets the scale by comparison with lattice QCD (or, in full QCD, experiment). Note that the running coupling remains finite even when the parameter $\alpha$ is sent to infinity. The orange bands in Fig. 4 indicate the onset of the decoupling solutions, which are close to the solutions seen on the lattice: The ghost dressing is finite in the infrared and the three-gluon vertex has crossed zero but not very far.

Repeating these calculations for general values of $(\alpha, \beta)$, one can identify lines of constant physics along which the solutions are identical up to rescaling. This is shown in Fig. 5 and entails that $\alpha$ and $\beta$ recombine to two parameters $c_0$ and $c_1$, where $c_0$ only rescales the system and $c_1$ is the actual parameter that distinguishes the scaling and decoupling solutions. Therefore, the family of solutions is due to the presence of the mass term $\beta$, or in general $\Delta_0$, which is entirely nonperturbative. The bending of the lines implies that without such a term ($\beta \to 0$) only the scaling solution would survive (as in Scenario A); if this term is dynamical, one obtains the family of decoupling solutions with the scaling solution as its endpoint.

## 3 Longitudinal singularities

Let us return to the question of longitudinal singularities. The condition $\Delta_L = -\Delta_0/Q^2$ requires purely longitudinal poles in either of the vertices appearing in the gluon DSE (Fig. 2). Such terms do not appear in the equations we solved so far: we employed tree-level tensors for the ghost-gluon, three-gluon and four-gluon vertices, which should be a good approximation for the transverse equation for $Z(Q^2)$ but cannot provide the full dynamics in the longitudinal sector. For example, the general ghost-gluon vertex depends on two tensors,

$$\Gamma_{\text{gh}}^{\mu}(p, Q) = -igf_{abc}\left[(1 + A)p^{\mu} + BQ^{\mu}\right], \tag{8}$$

where $p^{\mu}$ is the outgoing ghost momentum and $Q^{\mu}$ the incoming gluon momentum, and the two dressing functions $A$ and $B$ depend on $p^2$, $p \cdot Q$ and $Q^2$. The function $A$ is small in Landau gauge [23], which makes the tree-level tensor ($A = B = 0$) well-suited for approximations in the transverse sector. In turn, little is known about $B$ which only enters in the term $\Delta_L$.

To determine the longitudinal vertex dressing $B$, one must solve the DSE for ghost-gluon vertex shown in Fig. 5. This DSE can be read as an inhomogeneous Bethe-Salpeter equation

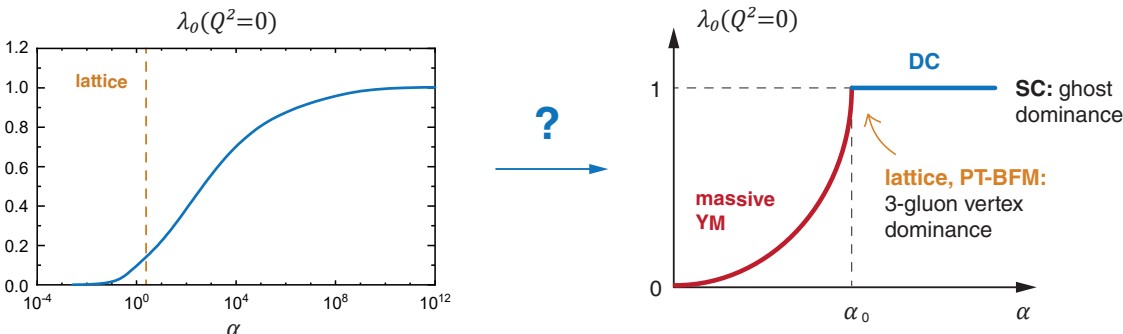

Figure 6: *Left:* Largest eigenvalue of the homogeneous BSE for the ghost-gluon vertex, which only for the scaling solution $\alpha \to \infty$ satisfies $\lambda_0 = 1$. *Right:* Sketch of a possible scenario with massless singularities in the vertices (blue) and the massive Yang-Mills regime without such singularities (red) [5].

(BSE) for $B$, and to determine if $B$ has a pole, one can equivalently solve the corresponding homogeneous BSE: If the lowest-lying eigenvalue $\lambda_0(Q^2)$ becomes 1 for some value of $Q^2$, the vertex must have a longitudinal pole. In particular, if $\lambda_0(Q^2 = 0) = 1$, the ghost-gluon vertex must have a *massless* longitudinal pole.

The BSE solution for the eigenvalue $\lambda_0(0)$ is plotted in the left of Fig. 6. One can see that $\lambda_0(0)$ indeed approaches 1 for $\alpha \to \infty$, which means that the ghost-gluon vertex *does* have a massless longitudinal pole, however only for the scaling solution. This would imply that only the scaling solution can satisfy the longitudinal condition (4) needed for gauge consistency.

Obviously this raises the question about the lattice decoupling solutions and the PT-BFM scheme, where longitudinal massless poles appear in the three-gluon vertex [35, 36]. In fact, it seems quite natural that if the ghost-gluon vertex does have such a pole, it would trigger longitudinal massless poles in all other correlation functions whose DSEs contain ghost loops, including the three-gluon vertex DSE in Fig. 2. In principle, the question could be settled by back-coupling the three-gluon vertex including its full momentum dependence and all its 14 Lorentz tensors, in which case one would arrive at coupled BSEs for the longitudinal sector of the ghost-gluon, three-gluon, four-gluon vertex etc. This leads to the situation sketched in the right panel of Fig. 6: The eigenvalue $\lambda_0(Q^2 = 0)$ would serve as an order parameter that distinguishes the massless (QCD-like) solutions from the massive Yang-Mills solutions, where the region close to the phase transition would be dominated by longitudinal poles in the three-gluon vertex and the scaling solution corresponds to a ghost dominance. If this picture were confirmed, it would indeed provide a strong indication that all QCD-like solutions are physically equivalent and simply generated by different mechanisms.

## 4 Conclusions

In this work we summarized a recent study of mass generation in Landau-gauge Yang-Mills theory. The corresponding Dyson-Schwinger equations admit a family of solutions characterized by two parameters. One of them only rescales the solutions, whereas the other distinguishes the range between a massive Yang-Mills-like regime on one side and the massless decoupling regime including the scaling solution on the other side. The existence of this family is tied to the term $\Delta_0$, which is nonperturbative and acts as an effective mass term in the equations.

For the Yang-Mills solutions, gauge consistency requires longitudinal massless poles in the vertices — or phrased differently, the existence of longitudinal massless poles triggers mass

generation in the transverse sector. We find that the ghost-gluon vertex indeed has such a pole, which establishes the consistency of the scaling solution. The consistency of the decoupling solutions requires longitudinal massless poles also in the three-gluon vertex (and possibly other ones), which is the observation in the PT-BFM scheme. In that case, the eigenvalue of the longitudinal Bethe-Salpeter equation would act as an order parameter that distinguishes the massless from the massive Yang-Mills regime. This can be tested in the future and if confirmed, it would provide further evidence for the decoupling solutions, with the scaling solution as their endpoint, being physically equivalent.

On a more practical note, calculations such as the one presented herein establish a first step towards *ab-initio* calculations of hadron properties with functional methods, which do not rely on any parameters except those in the QCD Lagrangian and whose only approximations amount to neglecting higher $n$-point functions, which makes them systematically improvable. A recent example is the calculation of the glueball spectrum in Yang-Mills theory, which is in agreement with lattice QCD calculations [37,38]. An important goal for future studies will be the extension of such calculations towards full QCD.

# Acknowledgements

G.E. would like to thank V. Petrov and the organizers for the invitation to give this talk at the XXXIII International Workshop on High Energy Physics. We thank João M. Silva for contributions in the initial stages of this project and Reinhard Alkofer, Christian Fischer, Markus Huber, Axel Maas and Joannis Papavassiliou for fruitful discussions. This work was supported by the FCT Investigator Grant IF/00898/2015, the Advance Computing Grant CPCA/A0/7291/2020, by EMMI, and by the BMBF grant 05P18VHFCA. It is part of and supported by the DFG Collaborative Research Centre SFB 1225 (ISOQUANT) and the DFG under Germany's Excellence Strategy EXC - 2181/1 - 390900948 (the Heidelberg Excellence Cluster STRUCTURES).

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
