# Peer review of "Studying mass generation for gluons"

_SciPost Physics Proceedings, doi:SciPost Phys. Proc. 6, 018 (2022)_

## Round 1 · Referee Report · Anonymous · 2022-3-29

Report
The infrared behavior of the gluon and ghost propagators in covariant gauges is studied in the Dyson-Schwinger approach. Proper attention is given to the contributions to the gluon propagator caused by quadratic divergences and by longitudinal massless singularities coming from the vertex functions; these contributions are combined so that the tensor structure of the gluon polarization operator agrees with gauge invariance.
Two parameters associated with these contributions make it possible to discriminate between widely discussed scaling and decoupling solutions of the Dyson-Schwinger equations. Possible independence of all physical (gauge-invariant) quantities on the deep-infrared behavior of the gluon propagator advocated by the authors may be associated with nonphysical nature of these parameters giving a hope that the Schwinger-Dyson approach in QCD can give rise to a computation of physical observables from first principles involving no additional parameters.
Requested changes
In caption to Fig.2 difference between blue, green, empty and orange circles
should be indicated.

---

## Editorial Decision

published